# Quantifying Assistive Robustness Via the Natural-Adversarial Frontier

Jerry Zhi-Yang He[1], Zackory Erickson[2], Daniel S. Brown[3], and Anca D. Dragan[1]

[1]University of California Berkeley, [2]Carnegie Mellon University, [3]University of Utah,
{hzyjerry,anca}@berkeley.edu, dsbrown@cs.utah.edu, zerickso@cmu.edu

**Abstract:** Our ultimate goal is to build robust policies for robots that assist people. What makes this hard is that people can behave unexpectedly at test time, potentially interacting with the robot outside its training distribution and leading to failures. Even just *measuring* robustness is a challenge. Adversarial perturbations are the default, but they can paint the wrong picture: they can correspond to human motions that are unlikely to occur during natural interactions with people. A robot policy might fail under small *adversarial* perturbations but work under large *natural* perturbations. We propose that capturing robustness in these interactive settings requires constructing and analyzing *the entire natural-adversarial frontier*: the Pareto-frontier of human policies that are the best trade-offs between naturalness and low robot performance. We introduce RIGID, a method for constructing this frontier by training adversarial human policies that trade off between minimizing robot reward and acting human-like (as measured by a discriminator). On an Assistive Gym task, we use RIGID to analyze the performance of standard collaborative Reinforcement Learning, as well as the performance of existing methods meant to increase robustness. We also compare the frontier RIGID identifies with the failures identified in expert adversarial interaction, and with naturally-occurring failures during user interaction. Overall, we find evidence that RIGID can provide a meaningful measure of robustness predictive of deployment performance, and uncover failure cases in human-robot interaction that are difficult to find manually. https://ood-human.github.io

**Keywords:** assistive robots, safety, human-robot interaction, adversarial robustness

## 1 Introduction

Deploying assistive robots in healthcare facilities and homes hinges crucially on their robustness, reliability, and the assurance that they can safely interact with individuals of all ages, from children to older adults. The worst-case scenario would involve an errant robotic arm causing injuries. However, slight variations or unpredictable elements in the operational environment can influence the behavior of learned robotic policies. Consequently, we must thoroughly examine the robot's response not just to regular situations but also to potential scenarios where unexpected events may unfold.

The complexities of stress testing robot policies, however, can be exceptionally high. For instance, verifying the robustness of a dish-loading robot in all kitchen configurations is highly costly [1]. Verifying human-robot interaction applications, in comparison, is even more challenging due to the fact that the human partner performs independent actions, changing the state that the robot's policy responds to, and potentially inducing distribution shift.

The common approach to probing the policy's robustness — perturbing inputs $x$ to the model — would translate here to perturbing the human partner's actions. But this can paint a misleading picture of the robustness of an assistive policy: while in theory any human action is possible, in practice humans won't move in arbitrary ways. A robot policy might thus fail under small *adversarial* perturbations but work under quite large *natural* deviations in human motion.

We thus propose to measure robustness to *natural* variations in human motion. This begs the question of how to define "natural". We propose a generative approach: given a dataset of motions —

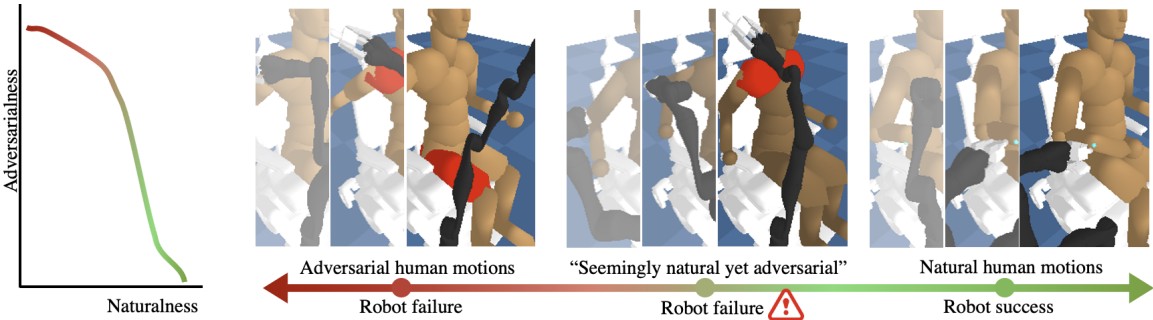

Figure 1: We propose measuring assistive robustness by considering the *naturalness* of human motions, and analyzing the entire natural-adversarial frontier: points with natural human motion that lead to good robot performance (left), unnatural motions that easily break the policy (right), and natural motions that still lead to failures (center). The full frontier paints a more useful picture of robustness than looking at a single point, and can be reduced to a scalar using the area under the curve (AUC): lower AUC is indicative of higher robustness.

demonstrated by the human in the target task or sampled from a trusted human model/simulator [2, 3] — we train a GAN [4, 5] and definite naturalness via the learned discriminator.

Next comes the question of *how* unnatural we allow human motions to be in our search for adversarial attacks against the robot: it is easy to break most policies if we allow motions that are very unnatural, and we might not find the interesting failure cases if we restrict the motion to be too close to what was seen in training. We thus advocate that robustness should not be assessed by looking at a single threshold, but rather by looking *at the entire natural-adversarial frontier*.

> *Our proposal is to measure assistive robustness by considering the naturalness of human motions and analyzing the entire natural-adversarial frontier.*

This Pareto frontier likely contains points of poor robot performance caused by unnatural human motions and points of good robot performance interacting with naturally moving humans. What is interesting lies in between: there exist natural motions that can lead to unexpected failures as in Fig. 1, leading to a high area under the curve.

We introduce RIGID, a method that constructs the natural-adversarial frontier by training adversarial human policies that trade-off between minimizing robot reward and acting in a human-like way, as measured by the discriminator [1]. We construct the frontiers for different policies — one trained to naively collaborate with the human, and others trained with robust Reinforcement Learning methods (e.g. by using populations) [6, 7]. While we find that prior robust RL methods improve policy robustness measured by RIGID, we are able to uncover edge cases where natural motions (natural with regard to our simulated human) cause robot failures. We then use RIGID to improve robustness by fine-tuning regular RL using data points identified by RIGID. Finally, we conduct a user study with naive users and an expert who attempts to lower the robot's reward. We find that RIGID can generate failure cases more effectively than manual effort, and is predictive of deployment performance.

## 2 Related Work

Adversarial attacks and robustness have long been studied in machine learning, with notable examples in computer vision [8, 9, 10, 11, 12, 13]. Researchers have successfully compromised state-of-the-art computer vision models with simple manipulations of pixels [14, 15] imperceptible to human eyes, which leads to real-world safety concerns. Since then, the community has persistently come up with many attacks [16, 17, 18] and defense mechanisms [19, 20, 11]. Most of the attack methods can be characterized by their allowed perturbation set, such as bounded l2-norm, and the method of optimizing adversarial examples [21]. Robustness methods, on the other hand, often require training against adversarial examples by adding them to the training data [9, 22, 23] as a way of performing robust optimization [24] or focus on regularization and smoothing techniques [25, 20, 26, 11].

Robust control [27] in robotics focuses on deriving safety guarantees in the presence of bounded modeling errors and disturbances [28, 29, 30, 31]. It has been applied to human-robot interaction

---

[1]While in our work we use a simulated human to generate data for the discriminator, in general, this data can also come from human-human or human-robot interaction in the task.

by allowing completely adversarial human motions (the full forward reachable set), as well as a restriction to only motions that are sufficiently likely under a human model [32]. Robustness in reinforcement learning is a growing field inspired by the brittleness of RL policies [33, 34]. Prior work on adversarial attacks focuses on single-agent [35] or two-player competitive settings [36, 37, 38, 39]. In collaborative settings, policies are known to be more brittle [2] than zero-sum games. Recent works have also looked at improving the generalizability in collaborative settings, which is a form of robustness against naturally occurring human distributions [6, 40, 41]. Our work is complementary, providing a way to measure robustness, and perhaps paving the way to new methods to improve it.

Dataset Aggregation (DAgger) [42] is a framework for passively incorporating failure cases into robot learning. One limitation of DAgger is that it requires online interaction with real humans in order to improve robustness – failing, and having the human correct or otherwise augment the data and retraining. This is undesirable in assistance and may cause harm to users. In contrast, our framework is an instance of active learning [43] that looks for failure cases of the robot policy before deployment. This way, we actively discover failure cases in simulation without real-world consequences.

Generative adversarial imitation learning (GAIL) [5] is a promising method for producing motions that mimic a set of demonstrations by training a discriminator on a dataset to provide the reward for a reinforcement learning agent. Researchers have experimented with variants of GAIL and found that it leads to high-quality motions in graphics [44], locomotion [45], and even controllable natural motions [46]. In this work, we build on the existing techniques for training GAIL to generate human motions that are perceived as natural. This allows us to study assistive robustness: by optimizing for the joint objective of being both natural and adversarial, we search for a range of plausible and typical human motions that create catastrophic failures for robot policies in assistive settings.

## 3   Natural-Adversarial Robustness

In this section, we first introduce the robotic assistance problem in Fig. 2, where the goal is to assist a human in a partially observable setting — the human may have hidden information, such as intent or preferences, that needs to be inferred on the fly. We then formulate the problem of adversarial but natural human motion, and reduce it to the optimization of an objective trading off between the two.

**Assistance as a Two-Player Dec-POMDP**. Following prior work [47, 48, 6] we model an assistive task as a two-agent, finite horizon decentralized partially-observable Markov decision process (Dec-POMDP), defined by the tuple $\langle S, \alpha, A_R, A_H, \mathcal{T}, \Omega_R, \Omega_H, O, R \rangle$. Here $S$ is the state space and $A_R$ and $A_H$ are the human's and the robot's action spaces, respectively. The human and the robot share a real-valued reward function $R : S \times A_R \times A_H \to \mathbb{R}$; however, we assume that the reward function is not fully observed by the robot, i.e., some of the reward function parameters (e.g., the specific goal location or objective of an assistive task) are in the hidden part of the state. $\mathcal{T}$ is the transition function where $\mathcal{T}(s' \mid s, a_R, a_H)$ is the probability of transitioning from state $s$ to state $s'$ given $a_R \in A_R$ and $a_H \in A_H$, $\Omega_R$ and $\Omega_H$ are the sets of observations for the robot and human, respectively, and $O : S \times A_R \times A_H \to \Omega_R \times \Omega_H$ represents the observation probabilities. We denote the horizon of the task by $T$.

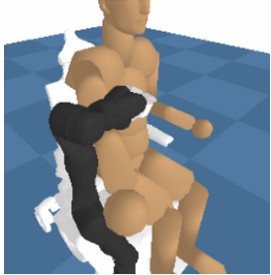

Figure 2: Assistive task: itch scratching with unknown itch locations. See appendix Sec. B for more details.

**Adversarial Human Motions.** Let a human trajectory denote a sequence of states and actions $\tau = (s_1, a_1^H, ..., s_T, a_T^H)$. and let function $\pi_H$ be the human policy that maps from local histories of observations $\mathbf{o}_t^H = (o_1^H, \ldots, o_t^H)$ over $\Omega_H$ to a probability distribution over actions. We define for an arbitrary human policy $\pi_H$, its corresponding assistive robot policy $\pi_{R(H)}$, which is the robot policy that is optimized to collaborate best with $\pi_H$ with respect to the joint reward $R$: $\pi_{R(H)} = \arg\max_{\pi_r} [\sum_t R(\pi_R, \pi_H)]$.[2]

We can then define an adversarial human policy $\tilde{\pi}_H$ with respect to an assistive robot policy. The policy $\tilde{\pi}_H$ minimizes the overall performance under the constraint that $\tilde{\pi}_H$ is similar to the original policy $\pi_H$[3] as measured by an $f$-divergence measure $D_f(\tilde{\pi}_H || \pi_H) \leq \delta$, and adjustable coefficient $\delta$

---

[2]For the sake of simplicity, we use a modified $R$ to denote the expected reward of the rollouts of human-robot policy pairs as $R(\pi_R, \pi_H) := \mathbb{E}_{a_r^t \sim \pi_r(s^t)} \mathbb{E}_{a_h^t \sim \pi_H(s^t)} \mathbb{E}_{s_1} [\sum_t R(s^t, a_r^t, a_h^t)]$.

[3]Typically, robot policies are trained via RL as a best response to a human model [2]; offline RL [49] is also possible and bypasses a human model, in which case the discriminator can be trained directly on the offline data.

controls the allowable perturbation set in which the adversarial human policy $\tilde{\pi}_H$ deviates from $\pi_H$, similar to adversarial perturbations in computer vision [11]:

$$\tilde{\pi}_H(\pi_{R(H)}, d) = \arg\min_{\pi'_H}[R(\pi_{R(H)}, \pi'_H)] \quad \text{s.t.} \quad D_f(\pi'_H || \pi_H) \le \delta \tag{1}$$

$\tilde{\pi}_H$ and $\pi_H$ are typically modeled by neural networks, making it difficult to directly compute their differences. We hereby introduce methods to compute their $f$-divergence via resulting trajectories.

**Approximating the Divergence** Measures such as KL divergence [50, 51, 3, 52], $\chi^2$ [53] divergence are commonly used to characterize the distance between two probability distributions. In the context of policy learning, this can be achieved by training a discriminator $\mathcal{D} : \tau \to \mathbb{R}$ to distinguish between trajectories sampled from $\tilde{\pi}_H$ and $\pi_H$. More specifically, $\mathcal{D}$ assigns a score $\mathcal{D}(\pi) := \mathbb{E}_{\tau \sim \pi}[\mathcal{D}(\tau)]$ to any policy $\pi$. It is trained to assign low scores to trajectories drawn from the true human policy $\pi_H$ and high scores to trajectories drawn from the adversarial policy $\tilde{\pi}_H$. We choose the LS-GAN objective [53] for training $\mathcal{D}$:

$$\mathcal{D} = \arg\min_{\mathcal{D}} \mathbb{E}_{\tau \sim \tilde{\pi}_H}\left[(\mathcal{D}(\tau) - 1)^2\right] + \mathbb{E}_{\tau \sim \pi_H}\left[(\exp\{\mathcal{D}(\tau) + 1)^2\right] \tag{2}$$

Previous work [53] has proven that this represents $\chi^2$ divergence when trained to optimality: $D_{\chi^2}(\tilde{\pi}_H || \pi_H) = \mathbb{E}_{\tau \sim \tilde{\pi}_H}[\mathcal{D}(\tau)^2]$. We illustrate this in appendix Sec. H. Note that while we focous on $\chi^2$ divergence in this paper, in practice one may use other divergence measures such as KL divergence and such as MMD (Maximum Mean Discrepancy) [54]. See appendix Sec. H for additional results.

**Adversarial Frontier** While previous work in Computer Vision (CV) [12, 11] assign the perturbation set to a fixed value, it is unclear what value we should assign to $\delta$ in the context of Human-Robot Interactions. Instead, we consider scanning over all possible values of $\delta$ to find the adversarial human policy under all different naturalness criteria. In other words, we consider all levels of "naturalness" in human motions. We consider the Lagrange dual function of the constrained optimization in Eq. (1):

$$L(\pi_R, \lambda) = \max_{\pi'_H} \left[ -R(\pi_R, \pi'_H) - \lambda \cdot D_f(\pi'_H || \pi_H) + \lambda \cdot \delta \right] \tag{3}$$

$$\tilde{\pi}_H(\pi_R, \lambda) = \arg\max_{\pi'_H} \left[ -R(\pi_R, \pi'_H) - \lambda \cdot D_f(\pi'_H || \pi_H) \right] \tag{4}$$

We have reduced the constrained optimization in Eq. (1) to an unconstrained optimization in Eq. (4) that balances "adversarialness" (playing an adversary to the robot by optimizing for $-R(\pi_R, \cdot)$) and "naturalness" (staying close to the canonical interaction by minimizing a divergence metric $\mathcal{D}(\cdot)$). The parameter $\lambda$ provides a knob that we can use to trade off how adversarial and how natural we would like $\tilde{\pi}_H$ to be. Setting $\lambda \to \infty$ results in a policy $\tilde{\pi}_H$ that closely resembles $\pi_H$ and yields a high reward. On the other hand, setting $\lambda = 0$ leads to a policy $\tilde{\pi}_H$ that is purely adversarial and causes harm to the assistive task by inverting the environment reward. By selecting the naturalness parameter $\lambda \in [0, \infty)$, we arrive at a spectrum of human motions that interpolate between adversarial and natural. This is meaningful in Human-Robot Interaction because while we may not need to worry about purely adversarial human behaviors, as they are less likely to happen in reality, we need to take precaution against human behavior that appears natural, yet causes robot failures.

Given these definitions, we can in principle compute the naturalness and adversarialness of all possible human motions, and plot them in a 2D coordinate system. This motivates the Natural-Adversarial curve — the Pareto frontier of non-dominated policies. These are policies that, compared to any other ones, have either a better adversarial score, or a better naturalness score, or both.

## 4 Computing the Natural-Adversarial Frontier

In this section, we propose a practical method that we call RIGID that can help us obtain the Adversarial Frontier in Sec. 3. Having motivated the concept of an adversarial human policy $\tilde{\pi}_H(\pi_R, \lambda)$ and the naturalness parameter, $\lambda$, in Sec. 3, we now discuss the Natural-Adversarial Pareto frontier (Sec. 4.1). We then introduce an algorithm, RIGID (Sec. 4.2), that efficiently approximates the natural-adversarial curve.

### 4.1 Connecting The Dots: the Natural-Adversarial Curve

**Points Along the Frontier:** To solve for Eq. (2), we start with sampled trajectories from $\pi_H$ (in our experiments, a human model; in general, this can also be data collected from human-human or

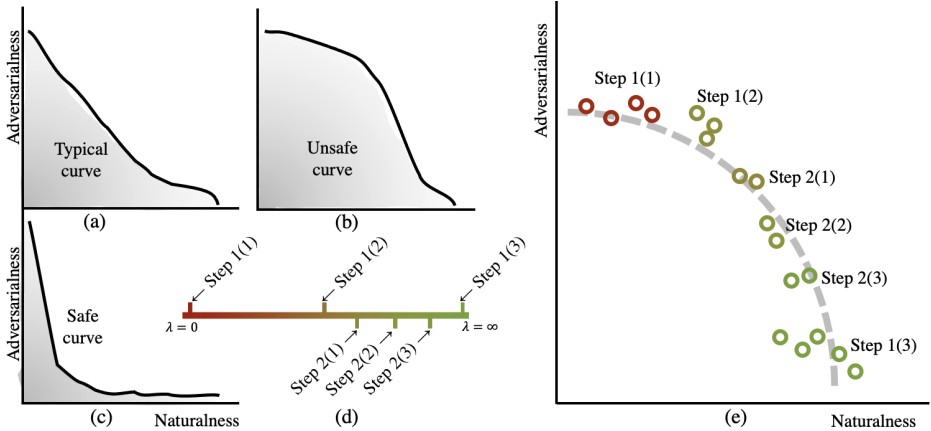

Figure 3: Left: different types of Natural-Adversarial curves. (b) has a higher AUC means it is less robust and more prone to unsafe behaviors. The ideal (c) curve looks like a sharp drop. Right: when running RIGID in Alg. 1, we first sample $\lambda$'s evenly in the log space over $\lambda \in [\lambda_{\min}, \lambda_{\max}]$ in (d). We then iteratively refine our picks by sampling more finely over selected intervals to gain better coverage of the curve in (e).

human-robot interaction). For a specific $\lambda$, we adapt GAIL [5] to our setting: we interleave training the adversarial policy $\tilde{\pi}_H$ and the discriminator $\mathcal{D}$ by iteratively (a) doing one step of policy optimization on Eq. (4) to adapt the policy and (b) doing one gradient step on Eq. (2) to adapt the discriminator. In practice, we use PPO [55, 56] as our policy optimization algorithm, and we use LS-GAN [53] with noise annealing to ensure the stability of the training process.

To plot the resulting $\tilde{\pi}_H$ in the natural-adversarial coordinate, we compute naturalness (x coordinate) by sampling trajectories $\tau \sim \tilde{\pi}_H$, and calculate the mean likelihood of $\mathcal{D}(\tau) < 0$. This is equivalent to having the discriminator classify the trajectories as from the original human model. We define adversarialness (y coordinate) as negative robot reward normalized to [0, 1]. See appendix Sec. D.

**Approximating the Curve:** Once we sample a sufficient number of $\lambda$s (see Sec. 4.2 on how to do this) from $[0, \infty)$, we connect the outermost points. The resulting curve characterizes the frontier of adversarial human motions under different levels of naturalness constraints (Eq. (4)) — or different perturbation sets (Eq. (1)). Intuitively, under the same naturalness value, lower adversarialness corresponds to more robust robot policies. Fig. 3 shows three different hypothetical curves, visualizing how the shape of the curve connects to the robustness of the policy.

**The AUC Score as a Scalar Robustness Metric:** Given a Natural-Adversarial curve, we compute the Area Under Curve (AUC) to quantitatively measure robustness as a single scalar. Lower AUC generally means stronger robustness, although there may be outlier human policies that can trigger unsafe behaviors, and a full plot of the curve is still necessary.

---

**Algorithm 1** RIGID

**Require:**
 1: Maximum refinements $d$, samples per round $k$
 2: Upper and lower bounds on $\lambda$: $\lambda_{\max}, \lambda_{\min}$
**procedure** RIGID ($\lambda_{\min}, \lambda_{\max}, d$)
 3: $\Lambda_{\text{all}} = [], S_{\text{nat}} = [], S_{\text{adv}} = []$
 4: $\lambda'_{\min} = \lambda_{\min}, \lambda'_{\max} = \lambda_{\max}$
 5: **for** $i = 1, \ldots, d$ **do**
 6: $\quad$ Find $k$ evenly spaced $e_1 \ldots e_k$ in $[\log \lambda'_{\min}, \log \lambda'_{\max}]$
 7: $\quad$ **for** $j = 1, \ldots, k$ **do**
 8: $\quad\quad \lambda_j = \exp(e_j)$
 9: $\quad\quad \Lambda_{\text{all}}.\text{append}(\lambda_j)$
10: $\quad\quad$ Compute $\tilde{\pi}_H$ using Eq. (4) and $\lambda_j$
11: $\quad\quad$ Compute $\text{nat}_j$ and $\text{adv}_j$ for $\tilde{\pi}_H(\lambda_j)$
12: $\quad\quad S_{\text{nat}}.\text{append}(\text{nat}_j), S_{\text{adv}}.\text{append}(\text{adv}_j)$
13: $\quad$ **end for**
14: $\quad \lambda'_{\min}, \lambda'_{\max} = \text{LARGEST-JUMP}(S_{\text{nat}}, \Lambda_{\text{all}})$
15: **end for**
16: **return** $S_{\text{nat}}, S_{\text{adv}}$
**end procedure**

---

## 4.2 Sampling Useful Trade-Offs

When we gradually increase $\lambda$, the resulting adversarial human motion often goes through mode changes, which lead to sudden increases in naturalness. Motivated by this, we need to select $\lambda \in [0, \infty)$ in a non-uniform manner, by procedurally expanding between the two $\lambda$'s where naturalness increases

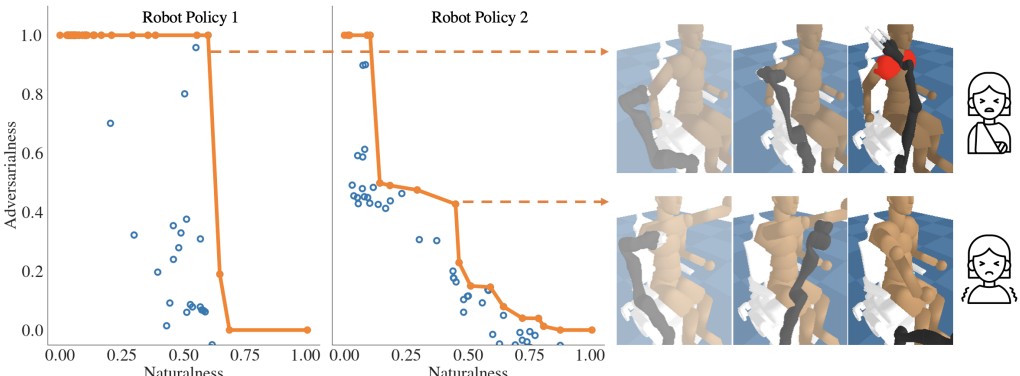

Figure 4: Natural-Adversarial curves of two different robotic policies trained using vanilla RL, with random seed differences. Every point corresponds to a motion policy found by RIGID for a particular $\lambda$ value. We highlight the frontier in orange. The visualization shows how points on the frontier correspond to failure cases.

the most. We do this through an iterative refinement approach by finding the largest gap in naturalness and increasing the sampling density there.

Our algorithm's pseudocode is shown in Alg. 1. The lists $S_{nat}$ and $S_{adv}$ keep track of the corresponding naturalness and adversarialness values of points along the Pareto frontier. We initialize $\lambda_{min} > 0$ to be the smallest $\lambda$ we consider, and $\lambda_{max}$ a sufficiently large number. We denote all the $\lambda$ considered so far as $\Lambda_{all}$. The procedure LARGEST-JUMP looks at all $\lambda$ selected so far and selects two $\lambda$ values between which the naturalness changed the most (see appendix Sec. A for the pseudo-code).

## 5   Experiments

### 5.1   How robust is Vanilla RL?

In this section, we use RIGID to analyze vanilla RL robot policies and to uncover possible natural (according to our simulator) motions that trigger failures. We then examine whether the RIGID frontiers are predictive of deployment performance: we test these policies with (a) end users and (b) an expert who adversarially attempts to get the robot to fail. We then plot the resulting behaviors relative to the RIGID-computed frontier. We find that RIGID is able to find points that are just as natural, but more adversarial[4]. Next, we analyze existing algorithms for robustifying RL policies and compare them according to RIGID. We find that RIGID does help differentiate between more and less robust policies. We also show that training with the natural adversarial human motions identified by RIGID leads to more robust RL policies. We leave implementation details, computational complexity, and data requirements to Appendix Sec. C.

**The Assistive Itch-Scratching Task** [57] is visualized in Fig. 2. The human is seated in a natural pose with a seat-mounted robot arm. Only the human knows the position of the itch location on their arm. Both agents are rewarded for scratching the itch location and penalized for contacting anywhere else on the human body. Success is achieved if the robot achieves meaningful contact for 25 timesteps.

First, we study whether assistive robot policies trained with SOTA reinforcement learning algorithms are robust. To do this, we first generate synthetic human policies by jointly optimizing for human and robot policies following [58]. We keep and freeze the resulting human policy as the synthetic human that we use, and train an assistive robot policy using the PPO algorithm to assist the synthetic human. The resulting robot policies achieve nearly 100% success rate with the training human model. We then use RIGID to come up with natural and adversarial human policies to attack this robot policy. See appendix Sec. D for further details.

In Fig. 4, we visualize the resulting human policies from the RIGID algorithm as points and connect the dots on the outermost boundary to form a Natural-Adversarial frontier. Visualizations of points with naturalness values in the range $[0.4, 0.8]$ show abundant failure cases — the adversarial human policy moves naturally (according to the model), yet causes the robot to fail. We show two examples in Fig. 4 with the dashed arrow. See appendix Sec. D for more visualizations.

---

[4]The caveat is that naturalness here is only as good as the synthetic human models we used in the experiments. This could be improved by using real-world human-human or human-robot interaction data

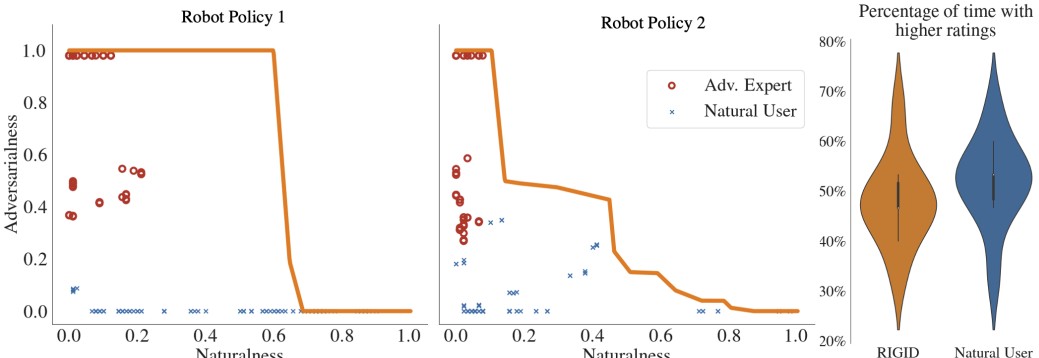

Figure 5: The RIGID-identified frontier for two policies. We find that RIGID pareto-dominates natural user and adversarial expert behaviors. When taking two behaviors with the same naturalness score — a user's and RIGID's — RIGID finds more adversarial solutions. On the right, we see that users indeed do not find a large difference between these two behaviors in "naturalness" (similarity to human training data).

While we can verify that points on the Natural-Adversarial curve are indeed failure cases of the robot policy, one may still question whether this curve is *exhaustive* and *faithful*. Here, *exhaustive* means that this curve encloses all the possible failure cases that one can produce, and *faithful* means that the trajectories deemed natural by the curve are judged as natural to the same extent by human beings.

**Exhaustiveness** We perform a two-part user study based on a virtual reality (VR) assistive gym plugin [59] to verify *exhaustiveness*. First, we recruit eight novice users who do not have prior experience interacting with the robot in VR. We inform them about the task, the objective and physical constraints and have them watch a few episodes of successful interactions. The robot then assists the users, simulating regular daily deployment conditions around normal users. Second, we stress-test the system to emulate what might happen under a wider range of users and over prolonged deployment. We have an expert (one of the authors) act adversarially to cause the robot to fail (collide). If the curve produced by the RIGID algorithm is *exhaustive*, trajectories from both the regular users as well as the expert should lie underneath the curve, which is indeed the case (Fig. 5): *points on the RIGID curve Pareto-dominate benign user and even expert adversarial data.* These results suggest that RIGID is more effective at finding edge cases than manual efforts.

We also see that for these two policies, the AUC is smaller for the second, so we expect it to be more robust. Anecdotally, the expert reported that the first policy was easier to break by simply extending their arm, which sent the policy into an unstable rotating behavior. This suggests that *RIGID correctly identified which policy is more robust*.

**Faithfulness** To test whether the curves are *faithful*, we ask 15 novice users to judge whether trajectory pairs of similar "naturalness" generated by RIGID and the previous users are qualitatively similar. We show users trajectories from the original human-robot pair. We then ask them to compare RIGID and user trajectories and rate which motion is closer to the reference data. The plot on the right of Fig. 5 confirms that the "naturalness" metric in RIGID corresponds to the user judgments.

## 5.2 Do robust RL methods improve robustness? Can training with RIGID examples help?

In this section, we leverage RIGID to efficiently find Natural-Adversarial curves for different robust robot learning methods, allowing us to compare their assistive robustness.

In Fig. 5.2, we compare the following methods: (1) Gleeve et al [7] use human-policy randomization during the co-optimization phase to improve the robustness of the robot, (2) PALM [6] leverages a distribution of different human behaviors to learn a latent representation that enables better generalizations. (3) We also study an oracle method called Robust GT, where we fine-tune Vanilla RL on failure cases from the Natural-Adversarial curve, by adding the failure cases to the training human population, and resuming training from the Vanilla RL checkpoint.

We notice that while existing robustness methods improve over Vanilla RL in terms of Assistive Robustness AUC, they have visible failure cases (see appendix Sec. F). We also evaluate the performance of different robust policies on assisting humans. The robust optimization method from Gleeve et al [7] improves AUC at the cost of worse performance when assisting more natural human policies. For PALM [6], we also observe this performance-robustness trade-off. We find that Robust

GT, which is trained on adversarial human policies found from RIGID achieves the best AUC reduction while also achieving good performance when assisting natural human policies.

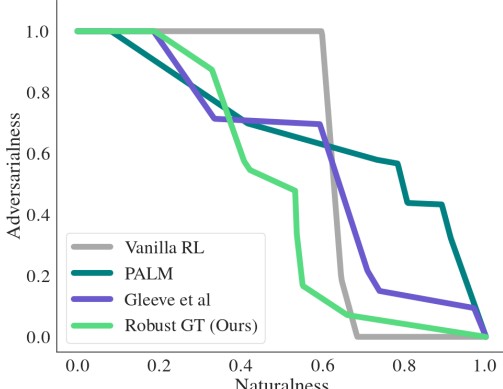

| Method | AUC ($\downarrow$) | Success ($\uparrow$) |
|---|---|---|
| Vanilla RL | 0.630 | 1.0 |
| Gleeve et al [7] | 0.584 | 0.83 |
| PALM [6] | 0.668 | 0.95 |
| Robust GT (Ours) | **0.473** | **1.0** |

Figure 6: Left: Natural-Adversarial curves of different methods. Right: We compute the Area Under Curve (AUC) of the Natural-Adversarial curves and the success rate of robot policies trained with different methods.

## 6 Limitations

One limitation is that the Natural-Adversarial curve carries some randomness. While the frontier exists, in practice one has to rely on multi-objective optimizations which have inherent randomness and sub-optimality. As a result, we can only approximate the frontier. Performing RIGID for more iterations under a diverse set of Naturalness objectives can help reduce this error.

The second limitation is that in our experiments, we rely on synthetic humans generated by human-robot co-optimizations [58, 57]. While such human agents perform reasonable movements (see visualizations in appendix Sec. B) and we have conducted extensive user studies in Sec. 5.1, future work should incorporate real-world human data to create more human-like simulated agents using Behavior Cloning or Offline RL [49]. It is also worth noting that in our studies, we use data from simulated humans as the canonical datasets on which we train GAN and measure "naturalness". While this may seem limiting, as we have only tested on our simulated humans, the framework is applicable to human-robot applications where there typically exists a canonical way of interaction (i.e. dressing requires the human to extend their arm to initiate the contact).

A third limitation is that in our implementation of Eq. (2), we use a memory-less discriminator. Because of this, the adversarial behaviors we find are primarily state-based. Using temporal naturalness measure can help us discover more subtle temporal adversarial behaviors. Training such temporal measures, however, imposes new challenges on policy learning and is beyond the scope of our focus.

Last but not least, our proposed algorithm RIGID relies on batch training of human adversarial policies. On the one hand, this means that in environments where policy learning is difficult (i.e. sparse reward), our framework is not applicable. On the other hand, computational complexity can be challenging. RIGID only demonstrates the feasibility of computing the Natural-Adversarial curve, and we believe that there are a number of ways to speed up RIGID. This includes Quality-Diversity methods [60, 61], using Evolutionary Methods to perturb one policy's rollouts, and fine-tuning one base adversarial policy instead of training all from scratch are promising directions.

## 7 Conclusions

We propose Assistive Robustness as a measure for evaluating the robustness of robot policies in assistive settings. Verified by user studies, we show that the Natural-Adversarial curve effectively represents to what extent the robot remains safe under plausible human perturbations. Our proposed algorithm RIGID is an initial attempt at effectively computing the curve. While we study physical interaction in healthcare as the main application in this work, we believe our framework can apply to general Human-Robot collaborative settings. We are excited for proposing a framework and tools to facilitate research safety in assistive and collaborative settings, as more sophisticated robotic applications are being deployed in the real world.

**Acknowledgments**

We would like to thank Cassidy Laidlaw and Erik Jones for helpful discussions on adversarial perturbations. This research was supported by the NSF National Robotics Initiative.

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

# Supplementary Material

## A  Algorithm

Below we provide the pseudo-code for the LARGST-JUMP procedure. Give a list of noisily increasing values — in our case, a list of noisily increasing "naturalness" values as we increase $\lambda$, the goal of LARGST-JUMP is to identify the two values, adjacent or not, that have the largest gap between them. This enables iterative refinement in Alg. 1.

---

**Algorithm 2** Procedure: LARGEST-JUMP

---

**Require:**
1: all $\lambda$'s $\Lambda_{\text{all}}$, all target values $S$.
2: length of sliding window $L$
**procedure** LARGEST-JUMP ($S$, $\Lambda_{\text{all}}$)
3:   Sort $S$ based on $\Lambda_{\text{all}}$.
4:   `lower_bound_so_far, upper_bound_so_far = [], []`
5:   `lower_idx_so_far, upper_idx_sov_far = [], []`
6:   `high_acc, low_acc = 10, -1`
7:   `high_idx, low_idx = len(S), 1`
8:   **for** $\text{i} = 0, \ldots, \text{len}(S) - 1$ **do**
9:     **if** $i = 1$ **then**
10:       `lower_bound_so_far.append(`$S$`[0])`
11:       `lower_idx_so_far.append(low_idx)`
12:     **else**
13:       `istart = max(i + 1 - `$L$`, 0)`
14:       `lower_bound_so_far.append(min(`$S$`[istart: i+1]))`
15:       `lower_idx_so_far.append(argmin(`$S$`[istart: i+1])) + istart)`
16:     **end if**
17:   **end for**
18:   **for** $\text{i} = \text{len}(S) - 1, \ldots, 0$ **do**
19:     **if** $i = 1$ **then**
20:       `upper_bound_so_far.append(`$S$`[-1])`
21:       `upper_idx_so_far.append(high_idx)`
22:     **else**
23:       `upper_bound_so_far.append(max(`$S$`[i: i+`$L$`]))`
24:       `upper_idx_so_far.append(argmax(`$S$`[i: i+`$L$`])) + i)`
25:     **end if**
26:   **end for**
27:   `max_gap_so_far = -1, max_gap_idxs = null`
28:   **for** $\text{i} = \text{len}(S), \ldots, 1$ **do**
29:     `diff = upper_bound_so_far[i] - lower_bound_so_far[i]`
30:     **if** `diff > max_gap_so_far` **then**
31:       `max_gap_so_far = diff`
32:       `max_gap_idxs = (lower_idx_so_far[i], upper_idx_so_far[i])`
33:     **end if**
34:   **end for**
      **return** $\Lambda_{\text{all}}$ `[max_gap_idxs[0]]`, $\Lambda_{\text{all}}$ `[max_gap_idxs[1]]`
**end procedure**

---

## B  Additional Details of the Environment Setup

**Assistive Gym Itch Scratching** We use the itch scratching environment proposed in [57] with the original settings. The biggest difference is that ***we limit the itch positions to randomizing from two fixed points, one in the middle of the forearm and one in the middle of the upper arm***, as opposed to freely sampling from any position on the arms. This is to simplify the robot assistance problem so that we can focus on studying robot robustness.

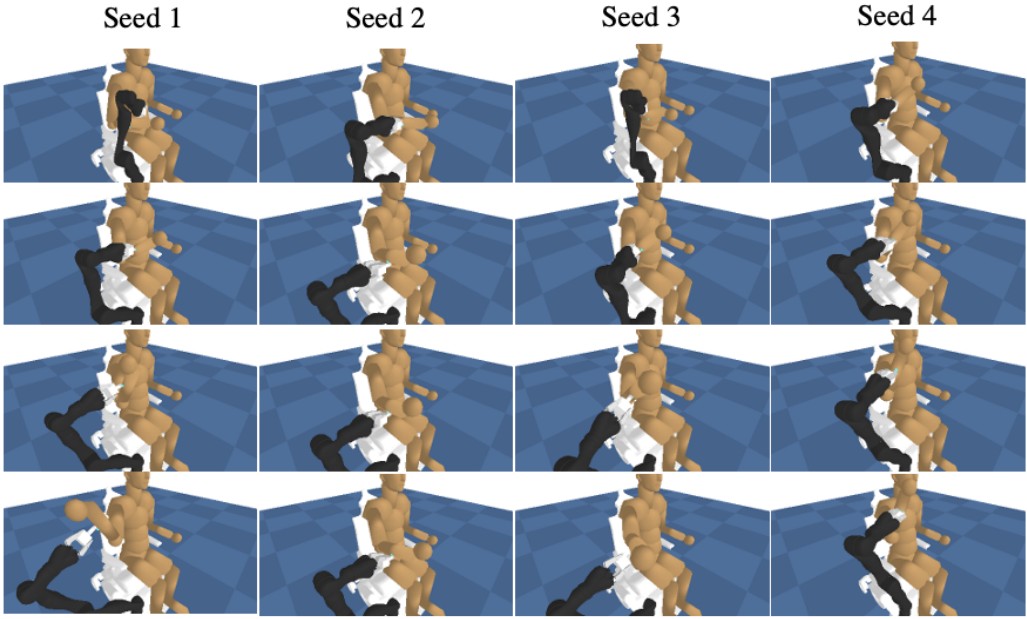

Figure 7: Visualizations of the trajectories of four different co-optimized human-robot policy pairs.

We also modify the environment time-span to 100 steps so as to speed up downstream RL training. The reward function in the original itch-scratching environment does not fully capture unwanted behaviors such as the robot swinging its arm and making unwanted contacts. We modify the environment reward function to increase the distance penalty (the robot's end-effect being far from the human) and add in contact penalty when the robot impacts areas other than the human's arms.

**Co-Optimization** We adopt the co-optimization framework proposed in [58], where we have both the human and robot jointly optimize for the task reward. We train both policies using PPO [55]. At every RL step, we update both the robot and the human policies.

More specifically, we use the ray library and train co-optimized policy pairs using batch sizes of 19,200 timesteps per iteration. We use the default learning rate and PPO hyperparameters in the RLLib library and train for a total of 400 iterations.

**Training Personalized Robot Policies** We keep the human policy from the co-optimized pair as our synthetic human policies. We can then train a personalized robot policy to assist the synthetic human. This is akin to programming robotic agents to assist humans. The resulting robot policy – which we refer to as personalized policies – is the focus of the paper and the target on which we compute the Natural-Adversarial curves. Note that there are different methods to find personalized policies besides running Vanilla RL. We describe them in more detail in Sec. C.

**Collecting Canonical Datasets** We collect datasets of 40 trajectories of synthetic humans and personalized policies as canonical datasets, which we later use to train GAN to compute "naturalness". Basically, the GAN enforces that perturbation trajectories stay in the proximity of the canonical trajectories. *Here by "natural", we mean human motions that are indistinguishable from the canonical trajectories.* This notation can apply to general human-robot interaction settings because such applications typically assume canonical trajectories (i.e. default dressing motions), and allow humans to fluctuate within some range of motions.

**Visualizations** Here in Fig. 7 we provide visualizations of the trajectories of four different co-optimized human-robot policy pairs. We use the same hyperparameters except for random seeds. The human motions are different amongst different seeds, and remain within reasonable motion range.

## C  Additional Details of the Experiment Setup

In this section, we talk about the details of different methods for training personalized policies. We also detail the training of GAN for generating natural-adversarial human behaviors.

## C.1 Learning Robot Policies

**Vanilla RL** We use off-the-shelf library on PPO algorithm. To facilitate policy training, we use the original robot policy in the co-optimized human-robot pair as the expert to guide the training. More specifically, we query the expert policy for actions and compute Behavior Cloning loss on the robot policy. We set RL loss coefficient to 0.1 and BC loss coefficient to 1.

The robot policy is a 4-layer MLP with 100 hidden size. We use batch sizes of 9,600 timesteps per iteration and train for a total of 240 iterations. We set environment gamma as 0.09. We use a learning rate of 0.00005, and an eps of 0.0001. We also set the clip parameter as 0.3 in PPO. In each iteration, we perform 30 epochs of policy optimization, with 20 mini-batch each. We use clipped policy loss, clip gradient norm of robot policy by 20, and clip the value function by 10.

**PALM** We adopt the same setup as in [6]. To create a diverse human distribution, we vary the itch position to randomly sample from anywhere on the two arms. This leads to more diverse human movements. We use a recurrent history of 4 timesteps for PALM, and use a 4-layer recurrent VAE with 24 encoder hidden size, and 4 latent size to predict human motions.

**Gleeve et al** Based on [7], we diversify the human population from the co-optimization phase. During co-optimization, we jointly train 1 robot policy with 3 human policies initialized from different seeds. The resulting human policies are different from each other and naturally induces diversity in robot training. During the personalization phase, we train 1 robot personalized policy to simultaneously work with the 3 human policies from co-optimization.

**Robust GT** We perform Robust GT by first computing the Natural-Adversarial curve, and then manually select adversarial human policies that leads to the lowest robot reward given that the policy naturalness $\in [0.2, 0.8]$. We visualize these failure cases in Sec. F. To train with these adversarial humans, we sample them at 15% rate beside the original synthetic human. While one can train robot policy this way from scratch, we load the previous vanilla RL policy and continue training with this enhanced population. Robust GT can be viewed as a variant of DAgger [42] with simulation-generated failure cases, or as a form of automatic curriculum learning [62, 63].

## C.2 Improving GAN training

While GANs are knownly difficult to train, there exists a large number of practical tricks in improving GAN training. We find that the most helpful tricks are: adding noise with annealing. LS-GAN, gradient penalty, and applying different weights for expert and human data. We experimented with training discriminators on concatenated observation and action, and find that it does not lead to much change. Thus we end up using observation-only discriminators.

Because human joints move at different rates and scales, it is important to add different amounts of noise to different joint observations. We compute the joint movements from the existing dataset [59], and multiply each joint movement's standard deviation by a factor of 10. We then anneal this by a decay rate of 0.98. We apply a gradient penalty of 0.3. We also set the expert loss rate in GAN as 4, and the agent loss rate as 1. This helps prevent the discriminator from overfitting to the agent data distribution and collapsing early in the training.

## C.3 Computaional Complexity

To generate the adversarial curve in Sec. 5.1, we perform three scans over $\lambda$ space, each time launching 18 different RL training in parallel for human policy learning. The three scans can be parallelized, and in total we perform 54 policy learning. Within each learning process, we run the PPO algorithm for 120 iterations, 4800 steps per iteration. This results in total of 0.5M environment timesteps. On Intel Xeon Skylake 6130 @ 2.1 GHz CPU, each policy learning 12 CPU core hours. In total, each Natural-Adversarial uses 648 CPU core hours.

## C.4 Data Requirement

To train the LS-GAN as Naturalness Measure, we collect 40 canonical trajectories, each with 100 timesteps, which corresponds to 15 seconds of human-robot interactions. In total, we require 10 minutes of canonical demonstrations.

### C.5 How to Compute Natural-Adversarial Curve on a New Domain?

In order to compute the Natural-Adversarial Curve on a new domain, we suggest the following steps:

1. Determine the key features. In the itch-scratching task, we find that robot poses, velocities and contact forces are the key features that determine whether the trajectories are dangerous. The naturalness measure is defined by these key features.

2. Determine the size of the canonical dataset. One may steadily increase the size of the dataset, and visualize the Naturalness measure on OOD data as done in Fig. 12. We find that 4,000 timesteps is a reasonable size.

3. Experiment with $\lambda = 0$ to ensure that a purely adversarial human policy can be learned. Experiment with $\lambda = \infty$ to ensure that a natural human policy that stays close to the canonical dataset can be learned. This step ensures that policy learning is working properly.

4. Perform RIGID to scan over $\lambda$ and plot the Natural Adversarial Curve.38

## D Additional Details of the Natural-Adversarial Frontier

To perform the RIGID algorithm to find the Natural-Adversarial Frontier Curve, we sweep over $\lambda \in [0.00001, 10]$. We perform RIGID algorithm in an iterative refinement manner as in Alg. 1. We keep 3 separate RIGID histories over 3 different random seeds. During each iteration, we select 6 new $\lambda$'s for each seed. We terminate after three iterations. This results to a total of 54 RL runs per curve.

**Plotting in the Natural-Adversarial Coordiate** We set the naturalness (x value) of the resulting policies as the mean prediction result from the discriminator. To compute the adversarialness, we normalize the negative robot reward in [200, 1400] range, and clip values that exceed this range to the boundary. We find this range of manually performing different motions VR to find the mean negative reward values of natural motions as well as adversarial human behaviors that lead to failures.

**More Natural-Adversarial Curves** We visualize additional Natural-Adversarial curves of different Vanilla RL robot policies, trained on differently-seeded synthetic humans.

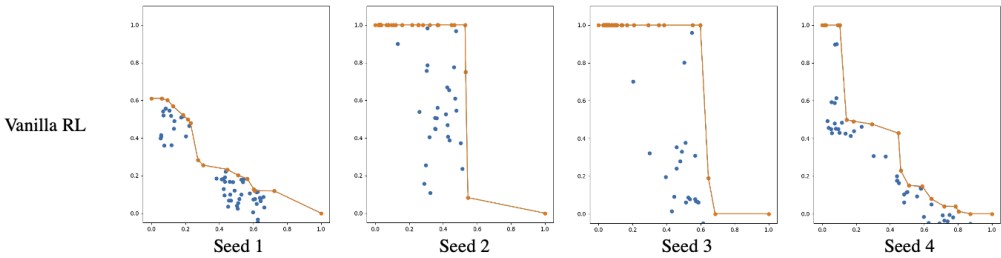

Figure 8: Visualizations of more Natural-Adversarial Curves

## E Additional User Study to Verify the Natural-Adversarial Curve

We perform two additional user studies to verify that (1) the trajectories deemed adversarial in the Natural-Adversarial Curve are indeed dangerous, and that (2) the trajectories deemed natural in the Natural-Adversarial Curve are indeed natural.

To study these hypotheses (1), we sample trajectory pairs from Fig. 5 where they are similar in x value (difference < 0.1), but different in y value (difference > 0.3). These correspond to trajectory pairs that are similarly natural, yet exhibit contrasting safety properties. We invite 5 novice users, familiarize them with the canonical trajectories, show them 30 trajectory pairs in randomized orders, and have them answer the following likert scale questions, and compute the final score by using the score of the second question minus the score of the first question. The higher the score is,

the more dangerous the resulting trajectory is. The result is plotted in the left part of Fig. 9.

| Questions for Hypothesis 1 |
|---|
| On a scale from 0 to 7, how well do you think the robot achieves the itch-scratching goal? |
| On a scale from 0 to 7, how well do you think the robot causes danger? |

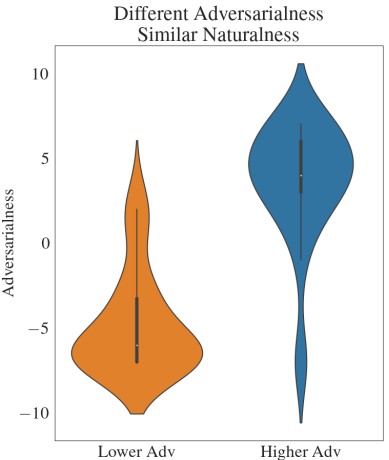 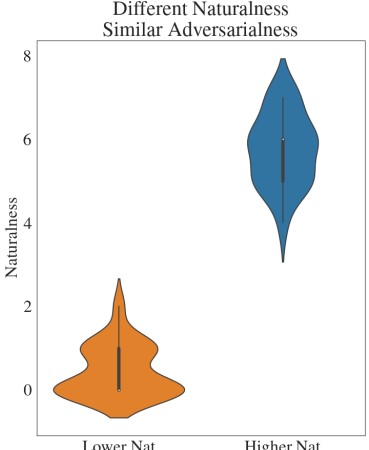

Figure 9: Additional user study results on verifying that (left) under similar naturalness, higher adversarialness in the plot corresponds to trajectories that are more dangerous, and (right) under similar adversarialness, higher naturalness in the plot corresponds to trajectories that are more natural

To study these hypotheses (2), we sample trajectory pairs from Fig. 5 where they are similar in y value (difference < 0.1), but different in x value (difference > 0.3). These correspond to trajectory pairs that are similarly safe, yet exhibit contrasting naturalness. We invite 5 novice users, familiarize them with the canonical trajectories, show them 30 trajectory pairs in randomized orders, and have them answer the following likert scale question. The higher the score is, the more dangerous the resulting trajectory is. The result is plotted in the left part of Fig. 9.

| Questions for Hypothesis 1 |
|---|
| On a scale from 0 to 7, how close do you think the trajectory resembles the canonical trajectory? |

# F  More Visualizations of Robot Failure Cases

In this section, we provide more visualizations of the failure cases in both keypoint trajectories and videos. We highlight the human body parts in red to indicate undesirable contact with the robot, such as the robot hitting the human's head.

**Trajectory Keypoint Visualizations** We visualize more failure cases of Vanilla RL as well as robust baselines in Fig. 10.

**Video Visualizations** We visualize more failure cases of Vanilla RL in videos in Fig. 11.

# G  Visualizations of the Discriminator

To gain more insight into the Naturalness discriminator, we selectively visualize three different discriminators learned under different $\lambda$ in Fig. 12. On the left we visualize the discriminators' accuracy on the canonical dataset, with increasing levels of noise added. We find that after applying the tricks in Sec. C.2, the discriminator is always able to correctly classify the canonical dataset. As we increase the $\lambda$ value, the resulting human trajectories (plotted under "OOD") becomes more

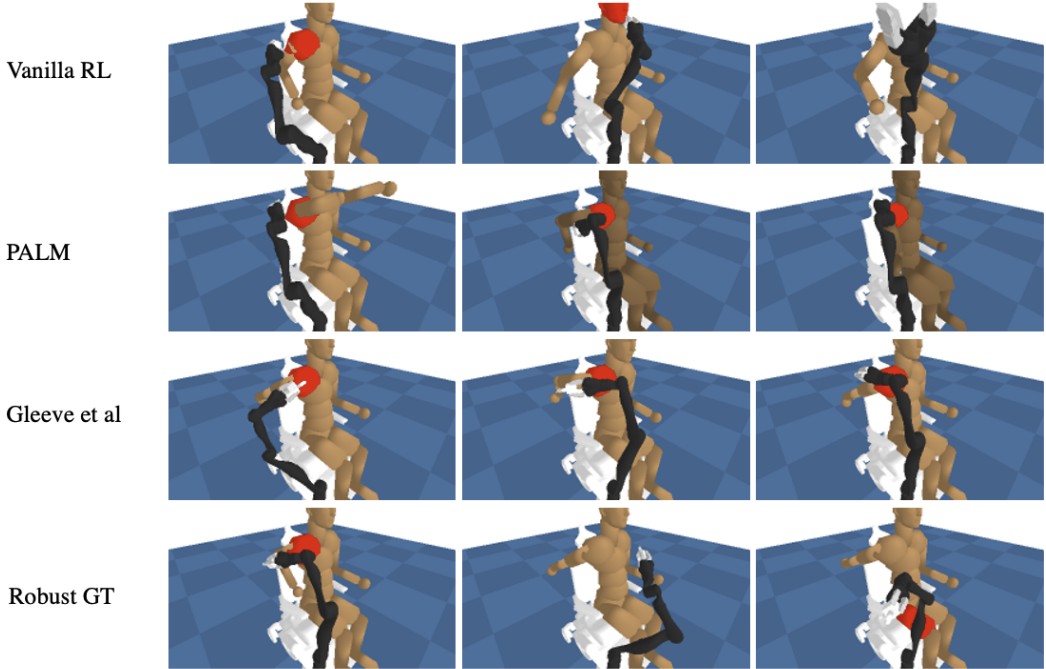

Figure 10: Visualization of keypoints of failure trajectories.

adversarial and easier to distinguish, which results in higher classification accuracies. On the right we visualize human poses under different noise levels, where the green ones are classified as positive (canonical) and the green ones are classified as negative by the discriminator..

## H  Different Choices of Naturalness Measures

As discussed in Sec. 3, we would like to constrain the adversarial policy $\tilde{\pi}_H$ to be similar to the original $\pi_H$ with respect to some $f$-divergence metric of the policy distribution. Commonly used divergence function include $\chi^2$ divergence, KL divergence, Wasserstein distance, etc. Such divergence measures are difficult to be estimated and optimized in Eq. (3). Thus, we present the variational form of them. The idea is that we can represent them using a discriminator optimized with a specially designed loss function. The advantage is that the discriminator is differentiable, compatible with Eq. (3), and is provably equivalent to the corresponding divergence measures when trained to optimality.

### H.1  Proof on Discriminator and $\chi^2$ Divergence

Based on [53], we train the discriminator $\mathcal{D}(\tau)$ and the policy $\tilde{\pi}(\tau)$ using the following loss.

$$\mathcal{D} = \arg\min_{\mathcal{D}} \frac{1}{2}\mathbb{E}_{\tau \sim \pi_H}\left[(\mathcal{D}(\tau) - b)^2\right] + \frac{1}{2}\mathbb{E}_{\tau \sim \tilde{\pi}_H}\left[(\exp\{\mathcal{D}(\tau) - a)^2\right] \tag{5}$$

$$\tilde{\pi}_H = \arg\min_{\pi} \frac{1}{2}\mathbb{E}_{\tau \sim \pi_H}\left[(\mathcal{D}(\tau) - c)^2\right] + \frac{1}{2}\mathbb{E}_{\tau \sim \pi}\left[(\exp\{\mathcal{D}(\tau) - c)^2\right] \tag{6}$$

Here $a, b, c$ are constants, $\pi_H$ is the canonical data distribution of the interaction, learned from human data or designed a-priori. Note that even though we work in the policy learning settings, $\mathcal{D}$ and $\tilde{\pi}_H$ are analogous to the discriminator and the generator in the GAN literature. $\mathcal{D}$ and $\tilde{\pi}_H$ are optimized iteratively till convergence. We hereby prove that the resulting $\tilde{\pi}_H$ minimizes the Pearson $\chi^2$ divergence. We first derive the optimal discriminator for a fixed $\pi$ in Eq. (5) as:

$$\mathcal{D}^*(\tau) = \frac{b\pi_H(\tau) + a\pi(\tau)}{\pi_H(\tau) + \pi(\tau)}$$

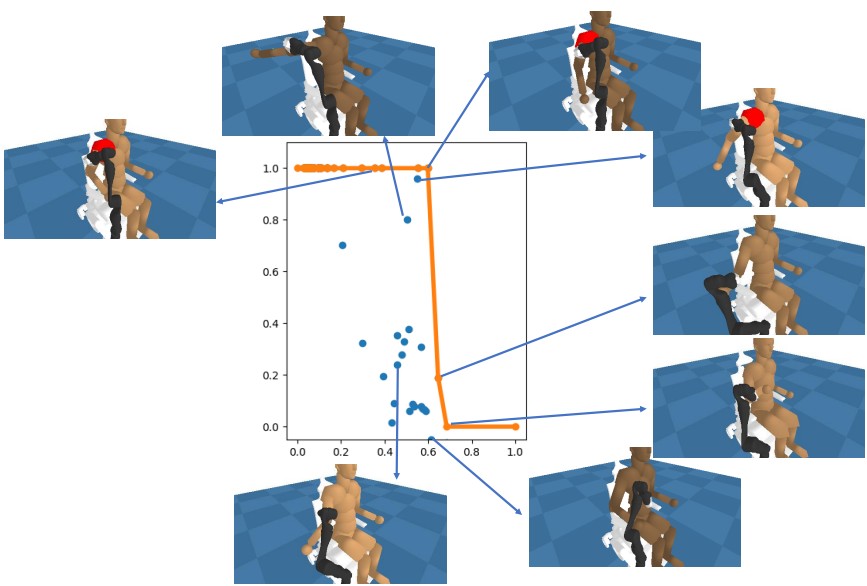

Figure 11: Visualization of videos of failure trajectories. **Click on the figure** to view the original image, where each trajectory contains a clickable link to its video.

We can then formulate the objective for $\tilde{\pi}$ in Eq. (6) as:

$$
\begin{aligned}
2\text{Loss}(\pi) &= \mathbb{E}_{\tau \sim \pi_H}\left[(\mathcal{D}(\tau) - c)^2\right] + \mathbb{E}_{\tau \sim \pi}\left[(\exp\{\mathcal{D}(\tau) - c)^2\right] \\
&= \mathbb{E}_{\tau \sim \pi_H}\left[\left(\frac{b\pi_H(\tau) + a\pi(\tau)}{\pi_H(\tau) + \pi(\tau)} - c\right)^2\right] + \mathbb{E}_{\tau \sim \pi}\left[(\exp\{\frac{b\pi_H(\tau) + a\pi(\tau)}{\pi_H(\tau) + \pi(\tau)} - c)^2\right] \\
&= \int_\tau \frac{((b-c)\pi_H(\tau) + (a-c)\pi(\tau))^2}{\pi_H(\tau) + \pi(\tau)}\mathrm{d}\tau, \quad \text{set } b - c = 1 \text{ and } b - a = 2 \\
&= \int_\tau \frac{(2\pi(\tau) - (\pi(\tau) + \pi_H(\tau)))^2}{\pi_H(\tau) + \pi(\tau)}\mathrm{d}\tau \\
&= \chi^2_{\text{Pearson}}(\pi_H + \pi \| 2\pi)
\end{aligned}
$$

This means that the optimal $\tilde{\pi}_H$ optimizes for the $\chi^2$ divergence. In practice, we select $b = 1, a = -1, c = 0$ so that

$$
\mathcal{D} = \arg\min_{\mathcal{D}} \mathbb{E}_{\tau \sim \tilde{\pi}_H}\left[(\mathcal{D}(\tau) - 1)^2\right] + \mathbb{E}_{\tau \sim \pi_H}\left[(\exp\{\mathcal{D}(\tau) + 1)^2\right] \tag{7}
$$

$$
\tilde{\pi}_H = \arg\min_{\pi} \frac{1}{2}\left[\mathcal{D}(\tau)^2\right] \tag{8}
$$

This is the formulation we used in Eq. (2). Q.E.D.

### H.2 Proof on Discriminator and KL Divergence

Based on [51, 3], we use the following objective for the discriminator $\mathcal{D}(\tau)$ and the policy $\tilde{\pi}(\tau)$:

$$
\mathcal{D} = \arg\min_{\mathcal{D}} \mathbb{E}_{\tau \sim \tilde{\pi}_H}\left[\log(1 + \exp\{-\mathcal{D}(\tau)\})\right] + \mathbb{E}_{\tau \sim \pi_H}\left[\log(1 + \exp\{\mathcal{D}(\tau)\})\right] \tag{9}
$$

$$
\tilde{\pi}(\tau) = \arg\min_{\pi} \mathbb{E}_{\tau \sim \pi}\left[\mathcal{D}(\tau)\right] \tag{10}
$$

We can rewrite Eq. (9) as:

$$
\mathcal{D} = \arg\min_{\mathcal{D}} \int \log\left(1 + \exp\{-\mathcal{D}(\tau)\}\right)\pi(\tau) + \log\left(1 + \exp\{\mathcal{D}(\tau)\}\right)\pi_H(\tau)\mathrm{d}\tau
$$

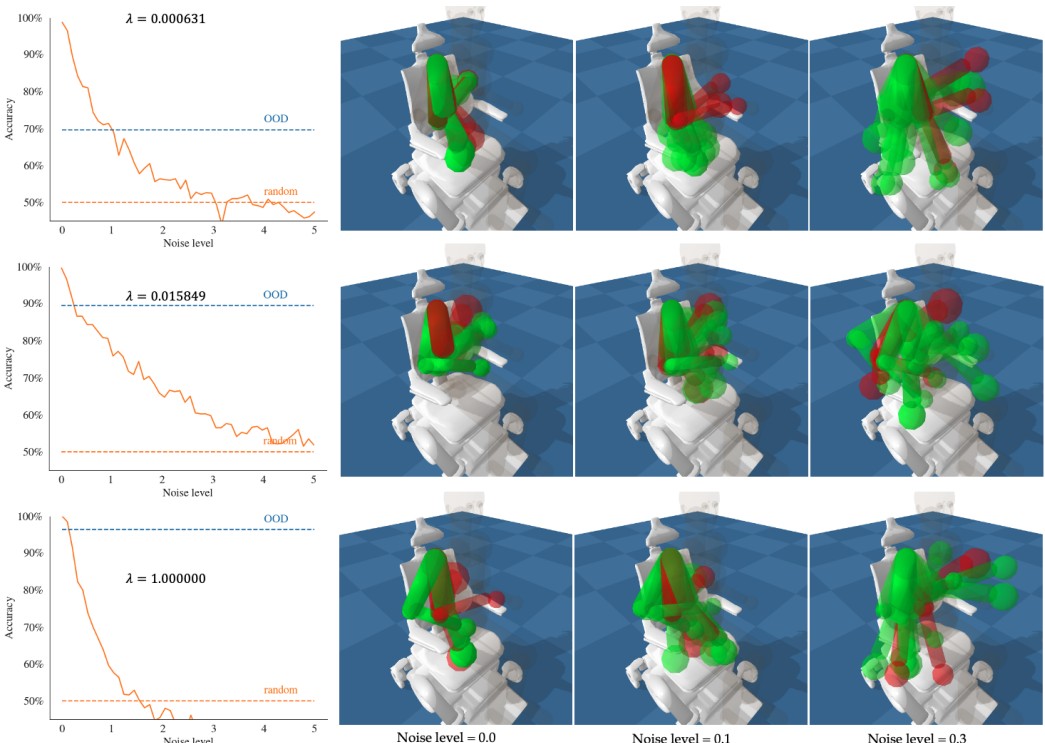

Figure 12: Visualizations of the discriminator. On the left we visualize the discriminators' accuracy on the canonical dataset, with increasing levels of noise added. The adversarial human trajectories are plotted under "OOD". On the right we visualize human poses under different noise levels, where the green ones are classified as positive (canonical) and the green ones are classified as negative by the discriminator.

The integral is minimized if and only if the integrand is minimized for all $\tau$, that is

$$\forall \tau, \mathcal{D} = \arg\min_{\mathcal{D}} \log\left(1 + \exp\{-\mathcal{D}(\tau)\}\right)\pi(\tau) + \log\left(1 + \exp\{\mathcal{D}(\tau)\}\right)\pi_H(\tau)$$

We can then show that the value for $\mathcal{D}(\tau)$ is $\mathcal{D}(\tau) = \log\left(\frac{\pi(\tau)}{\pi_H(\tau)}\right)$. Plugging this into the objective function in Eq. (10), we get

$$\text{Loss}(\pi) = D_{\text{KL}}(\pi(\tau)||\pi_H(\tau))$$

This means that the optimal $\tilde{\pi}_H$ optimizes for the KL divergence. Q.E.D.

## H.3 Discriminator-free Method: Maximum Mean Discrepancy

Maximum Mean Discrepancy (MMD) [54] is a kernel-based statistic test used to measure the difference between two distributions, and can be used as a loss/cost function in machine learning algorithms for density estimation.

Formally, given random variables $X, Y$, a feature map $\phi$ mapping $X$ to another feature space $\mathcal{F}$ such that $\phi(X) \in \mathcal{F}$, we can use the kernel trick to compute the inner product of $X, Y$ in $\mathcal{F}$ as $k(X, Y) = \langle\phi(X), \phi(Y)\rangle_{\mathcal{F}}$. We define feature means as a probability measure $P$ on $X$, which takes $\phi(X)$ and maps it to the means of every coordinate of $\phi(X)$:

$$\mu_P(\phi(X)) = \left[[\phi(X)_1], ..., [\phi(X)_m]\right]^T$$

The inner product of feature means of $X \sim P$ and $Y \sim Q$ can be written as $\langle\mu_P(\phi(X)), \mu_Q(\phi(Y))\rangle = \mathbb{E}_{P,Q}[\langle\phi(X), \phi(Y)\rangle_{\mathcal{F}}]$, Maximum Mean Discrepancy is defined as

$$\text{MMD}^2(P, Q) = ||\mu_P - \mu_Q||^2_{\mathcal{F}}$$

In practice, we can leverage MMD to measure the distance between $\tilde{\pi}_H$ and $\pi_H$, by collecting datasets of trajectories from each policy, and computing the MMD distance of the two sets. We can then

plug in MMD to replace $D_f$ in Eq. (4). This way, the divergence can be directly estimated without training a discriminator. We use the MMD implementation in [64], which uses a radial basis function as the kernel function.

We then conduct the main experiments using MMD distance on the same canonical human-robot policies in Sec. 5 to search for adversarial human policies. Note that we can apply the same RIGID framework to automatically scan for $\lambda$ values, and compute the natural-adversarial frontier. We find that MMD is similarly effective to using LS-GAN as the naturalness metric. For comparison, we take all the resulting adversarial human policies discovered by MMD and plot them under the same LS-GAN naturalness plot as in Fig. 5.

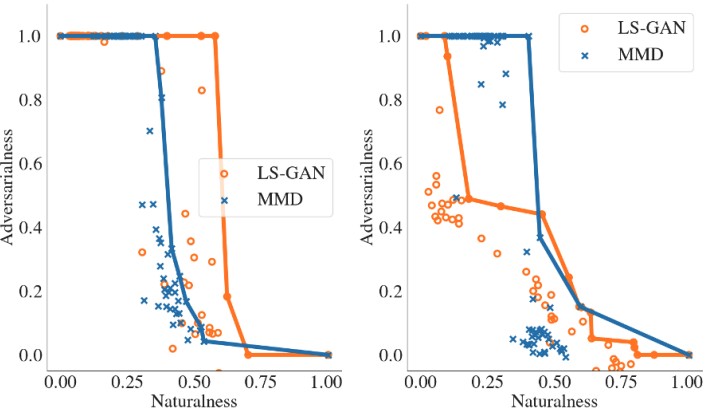

Figure 13: Natural Adversarial human policies found by MMD (blue) plotted alongside the ones found in the main paper (orange).

This shows that while MMD metric underperforms in LS-GAN in certain scenarios in terms of finding adversarial natural human policies, it is also able to find novel scenarios that are not discovered by by LS-GAN. Overall, this suggests that the Natural-Adversarial framework is applicable to different types of naturalness metrics, and using an ensemble of multiple metrics may outperform using a single metric.

# I  More details on the User Study Setup

We provide more details on the user study in Sec. 5.

## I.1  VR Visualizations

In the following figure Fig. 14, left and center are the user interacting with virtual robots through the HTC VIVE headset and the hand controller. The right is the first-person view in VR[59].

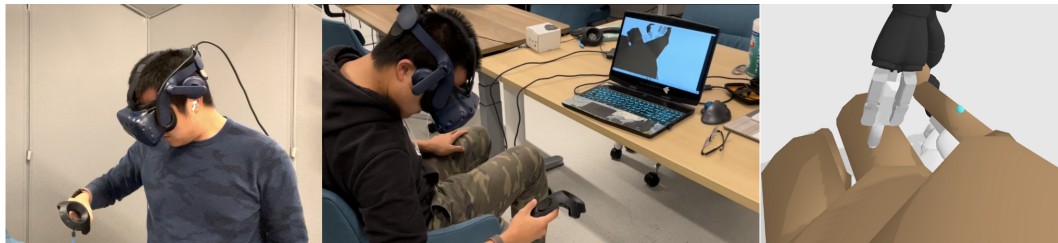

Figure 14: Workstation for performing the VR user study, where we have the human perform interaction with the robot in the calibrated VR environment.

We first have the users watch 3 iterations of canonical trajectories executed by the personalized robot policies and the original synthetic humans. We then instruct then to perform similar trajectories in their own ways.

## I.2 Questionaire GUI Interface

For the study on evaluating faithfulness, we use the following interface in Fig. 15, where we display a canonical trajectory on the left, and display single-timestep snapshots of two trajectories, one from RIGID policies and one from user VR executions. We ask the user to select which snapshot has stronger correspondence to the left trajectory. We randomize the sequence for each question.

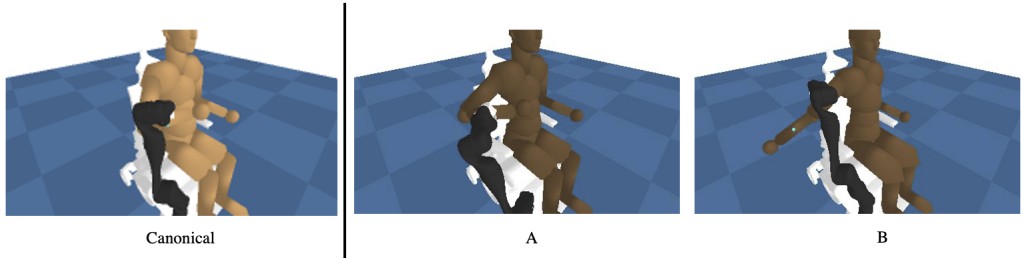

Figure 15

