# OpenReview forum: "Quantifying Assistive Robustness Via the Natural-Adversarial Frontier"
_robot-learning.org/CoRL/2023/Conference — CoRL 2023 Poster_

### Official Review · Reviewer_KeGh · 2023-07-17

**Confidence:** 4
**Originality:** Fair
**Technical Quality:** Good
**Clarity Of Presentation:** Very Good
**Impact:** 3

**Recommendation:**

Weak Accept: I recommend accepting the paper, but will not argue for my recommendation if the majority of other reviewers have a different opinion.

**Review:**

The proposed approach offers a means for incorporating naturalness into the evaluation criteria of assistive policies.  The authors provide an interesting perspective on the use of naturalness, and the use of models to quantify the performance with respect to how adversarial and natural the rollouts are.  The general principle of using a discriminator seems to be sound, and no concerns in general arise from this choice.  However, this immediately casts the problem as one of learning this discriminator, and so any concerns would be related to the amount of data available to reliably train such a discriminator.

There are several notable limitations, some of which were pointed out by the authors.  First, human movement was generated from synthetic data generated from a simulator.  Hence, there is still no guarantee that the discriminator can correctly learn a reasonable “naturalness” score.  The authors do not state under what conditions their approach may be applicable nor how generalizable their approach could be, nor discuss any concerns with sim-to-real transfer or how much data is needed from a real human if using human-derived data.

Second, the computational overhead of training on the order of 100 policies for evaluation to quantify the AUC seems overly prohibitive and contradicts the premise of practical evaluation of assistive policies.  Moreover, it is not clear how well the trained policies perform against other approaches that purely examine the adversarial evaluation of policies.  It would certainly help to see a comparison of the authors’ method against other approaches that deal only with the adversarial case via ablations and/or comparisons with existing work [9], [10].  It would be good to see both performance and training time reported in such a comparison.

Third, the result is heavily influenced by the discriminator, yet the authors do not evaluate the quality of the trained discriminator, nor do they properly evaluate its classification performance (e.g. using test datasets) and ability to detect out-of-distribution samples.

**Quality Of The Limitations Section:**

Limitations are addressed clearly

**Questions For Rebuttal:**

How can we be assured that the discriminator produces a sensible result for naturalness?

Could the authors please comment on the feasibility of using the approach to train robust policies given the substantial computational overhead required, and any practical limitations (environment complexity, complexity of the safety conditions, etc.)?

**Robotics Focus:**

Relevant but unlikely to deploy to hardware in near future

**Summary Of Paper:**

This paper proposes an approach to assessing the robustness of assistive policies in the presence of humans.  Typical approaches to robustness quantification are adversarial - purely focusing on the worst-case, without considering the realism of the failures.  This paper falls in the line of works seeking to produce robustness for “in-domain” or “in-distribution” failures (here, dealing with abnormal human poses).  The authors propose a solution that visualizes a Pareto frontier over two axes: the degree of “adversarialness” and the degree of “naturalness”, as the output of a learned discriminator the classifies the naturalness of the policy rollout.  Ideally, one would want a small AUC for the curve, meaning that there does not exist natural-behaving trajectories that are adversarial to the robustness of the system.

**Summary Of Recommendation:**

While the work is in general attacking an interesting problem, I do not feel there are sufficient differences from existing approaches that leverage out-of-distribution (OOD) examples for training, such as variants of DAgger and the like to warrant acceptance.  Moreover, the evaluation and discussion of various aspects, such as the discriminator, computational complexity, and practical utility are lacking.  I do not recommend acceptance as currently written.

Update: I have increased my overall score per the revised paper and the authors' rebuttal.

---

### Official Review · Reviewer_KfMg · 2023-07-18

**Confidence:** 5
**Originality:** Good
**Technical Quality:** Fair
**Clarity Of Presentation:** Fair
**Impact:** 2

**Recommendation:**

Weak Reject: I recommend rejecting the paper, but will not argue for my recommendation if the majority of other reviewers have a different opinion.

**Review:**

Significance and Originality
-------------------------
(+) The submission tackles an important question: How can we quantify robustness with respect to the natural variations within human behavior?

(+) The proposed measure seems novel.

(-) I have concerns regarding the meaningfulness of the computed measure. (see below under quality)

(-) due to high computational costs (multiple runs of reinforcement learning) further questions the usefulness.

Clarity
-------
(+) The paper is well-written and mostly clear.

(-) However, some important details are not mentioned (See questions).

(-) Eq. 4 seems wrong because the equality does not hold for a fixed discriminator. It needs to be stated as a saddle-point problem ($\underset{\pi}{\max}\underset{\mathcal{D}}{\min}$.

(-) The method is derived based on the reverse KL divergence using the binary cross entropy loss, however, Section 4.1 states that a LS-GAN is used. Why not derive the method based on the $\chi^2$ divergence?


Quality
-------
(-) The data points in the naturalness-adversarialness plots seem to be quite unreliable. When optimizing a policy to minimize a trade-off between task-reward reward and divergence to the human policy, all data points should be on the Pareto frontier (increasing $\lambda$ should result in higher naturalness and lower adversarial reward). However, based on Fig. 4, some data points are quite far from the frontier. The large noise can be caused both, by learning suboptimal policies, but also by not reliably estimating the divergence for computing the naturalness score. It would be interesting to compare the curves to one where the maximum mean discrepancy to the human demonstration is used for quantifying "naturalness".

(-) The Pareto frontier is created based on the samples that achieve largest adversarialness, which introduces a bias due to the noise in the data points, leading to a systematic overestimation of the adversarialness, given a certain level of naturalness. As a result, the results of the user study are not surprising.

(-) Judging from the footnote at page 3, it seems like all human actions and robot actions are sampled at the beginning of the episode and independently from each other. This would be a major limitation that is not well described.

Minor Comments
-----------------------
- I think (left) and (right) should be swapped in the caption of Fig.1
- line 43 "definite" should be "define"

References
----------
Gretton, A., Borgwardt, K. M., Rasch, M. J., Schölkopf, B., & Smola, A. (2012). A kernel two-sample test. The Journal of Machine Learning Research, 13(1), 723-773.

**Quality Of The Limitations Section:**

Additional details required

**Questions For Rebuttal:**

(-) The data points in the naturalness-adversarialness plots seem to be quite unreliable. When optimizing a policy to minimize a trade-off between task-reward reward and divergence to the human policy, all data points should be on the Pareto frontier (increasing $\lambda$ should result in higher naturalness and lower adversarial reward). However, based on Fig. 4, some data points are quite far from the frontier. The large noise can be caused both, by learning suboptimal policies, but also by not reliably estimating the divergence for computing the naturalness score. It would be interesting to compare the curves to one where the maximum mean discrepancy to the human demonstration is used for quantifying "naturalness".

(-) The Pareto frontier is created based on the samples that achieve largest adversarialness, which introduces a bias due to the noise in the data points, leading to a systematic overestimation of the adversarialness, given a certain level of naturalness. As a result, the results of the user study are not surprising.

(-) Judging from the footnote at page 3, it seems like all human actions and robot actions are sampled at the beginning of the episode and independently from each other. This would be a major limitation that is not well described.

**Robotics Focus:**

Highly relevant to robotics but no hardware experiments

**Summary Of Paper:**

The paper proposes a measure to quantify the robustness of a robot policy with respect to natural variations in the human behavior.
For measuring the robustness, first a robot policy is trained for given human trajectories. Then, several adversarial human policies are trained, that use GAIL to imitate the human trajectories, but also have an additional reward component to minimize the task reward of the previously learned robot policy. The different GAIL instances use different weights to trade off these two objective, leading to adversarial policies with different levels of naturalness (closeness to the human demonstrations). Robustness is then quantified as the area under the approximated curve that plots negative task reward over naturalness (smaller values indicate higher robustness). In a user study with 9 non-experts, the negated task reward is always below the approximated curve, based on the computed naturalness (using the GAIL discriminator) of the user behavior.

**Summary Of Recommendation:**

The paper tackles an important problem in HRI. However, I'm currently not convinced about the soundness, reliability and practicability of the proposed measure. I think that it needs to be derived based on the $\chi^2$ divergence, and Eq.4 should be stated as saddle-point problem. Furthermore, I think that more reliable estimates of the naturalness/divergence should be evaluated. It seems difficult to improve the practicability, but at least the limitations should be more clearly stated (regarding the optimization of the Dec-POMDP that involves a human actor).

---

### Official Review · Reviewer_wTf2 · 2023-07-20

**Confidence:** 2
**Originality:** Good
**Technical Quality:** Good
**Clarity Of Presentation:** Good
**Impact:** 2

**Recommendation:**

Weak Accept: I recommend accepting the paper, but will not argue for my recommendation if the majority of other reviewers have a different opinion.

**Review:**

# Comments

In this paper, the authors employ an adversarial learning approach to learning human policy for evaluating the control policies of assistive robots. The objective of this approach is to obstruct the robot's policies with natural movements. In this point, the proposed method is related to robust adversarial reinforcement learning (http://proceedings.mlr.press/v70/pinto17a.html). Therefore, we believe it's necessary to compile related research in this field and clarify its relation to our proposed method.


The method of assessing the robustness of robot control policies inherently incorporates policy learning into the evaluation process. Given that policy learning generally results in unstable performance, it is conceivable that this evaluation metric carries some randomness. What kind of problems might this present when utilizing this method, and how should the authors address these issues?

In order to efficiently evaluate the strategy using RIGID, I believe it's necessary to properly set the value of lambda. How should it be set for the best experimental results?

The experimental environment utilized is taken from the environment proposed in [55], hence the description of tasks has been omitted. For efficient comprehension of the experimental content, the basic task setting should be explained within this paper. Moreover, I believe it's crucial to effectively visualize the adversarial and natural movements acquired through adversarial learning. While there are snapshots in the appendix, discerning the naturalness and adversarial nature of the movements is challenging.


**Quality Of The Limitations Section:**

Additional details required

**Questions For Rebuttal:**

- Descriptions of the relationship between the approach that learns robust policies using adversarial learning and the proposed method
- Descriptions of the impact of learning instability on the evaluation
- Detailed descriptions of the experimental settings

**Robotics Focus:**

Highly relevant to robotics but no hardware experiments

**Summary Of Paper:**

In the domain of human motion assistive robots, this paper introduces a natural-adversarial frontier to assess the robustness of a robot's control policy against natural and adversarial human movements. The authors applied RIGID in a simulation environment and confirmed that their proposed method could generate more comprehensive adversarial movements than those created by humans. They also verified that the performance of robust polices can be compared using RIGID.

**Summary Of Recommendation:**

I believe that quantifying the robustness of the policies of assistive robots is valuable. On the other hand, there is a need to adequately justify the inclusion of a learning algorithm in the quantification.

---

### Official Review · Reviewer_XUqi · 2023-07-20

**Confidence:** 3
**Originality:** Good
**Technical Quality:** Good
**Clarity Of Presentation:** Good
**Impact:** 4

**Recommendation:**

Strong Accept: I recommend accepting the paper and will argue for my recommendation even if other reviewers hold a different opinion.

**Review:**

Pros
- **Quality & Clarity**
  - This paper is well-written and easy to follow. The experiments are well-designed and comprehensive.


- **Originality & Significance**
  - The idea of naturalness-adversarialness frontier is useful, novel, and well-motivated
  - The proposed adversarial training method that trades off naturalness and adversarialness based on GAIL is well-motivated.

Cons
- The setting of assistive tasks is not explained.
- The implementation of baselines is not explained (Section 5.3). For example, what is the implementation of RobustGT and PALM? It would be better to explain them briefly


**Quality Of The Limitations Section:**

Limitations are addressed clearly

**Questions For Rebuttal:**

1. In line 129, why do you need approximating KL divergence with discriminator?
2. In Section 5.3, how do you fine-tune vanilla RL?


**Robotics Focus:**

Highly relevant to robotics but no hardware experiments

**Summary Of Paper:**

This paper develops robust policies for robots that assist humans. However, this is challenging because measuring robustness is difficult since typical adversarial perturbations may not accurately represent natural human interactions. To address this, the authors propose constructing and analyzing the entire natural-adversarial frontier, representing human policies that balance naturalness and low robot performance. They introduce a method called RIGID, which trains adversarial human policies that trade off between minimizing robot reward and behaving human-like. Using RIGID on an Assistive Gym task, they evaluate the performance of standard collaborative reinforcement learning and existing robustness-increasing methods. The results suggest that RIGID can offer a meaningful measure of robustness, predicting deployment performance, and identifying challenging failure cases in human-robot interaction that are difficult to discover manually.

**Summary Of Recommendation:**

This paper identified a missing aspect, naturalness, in robust RL, and proposed a viable metric to quantify the performance of a policy's robustness under different naturalness. This aspect is important since in the actual use case of assistive robots, most perturbation from users is unlikely to be unnatural and only a few adversarial users can produce adversarial perturbation. Thus, to my knowledge, this is will be an important evaluation metric in the research of human-robot interaction.

---

### Decision · Program_Chairs · 2023-08-30

**Decision:**

Accept (Poster)

**Comment:**

The paper introduces an interesting metric for measuring and analysing the robustness and naturalness of HRI policies, by analysing the natural-adversarial frontier. Most reviewers have noted the originality of this work, and the main concerns revolved around the randomnesss of the introduced estimator, and clarifications regarding the presentation of the methodology. The authors have addressed adequately the concerned, and improved both the presentation of their method and their experimental results. After the rebuttal the paper was improved and is an interesting contribution for advancing safe learning in HRI.